# Cocoa Nanoparticles to Improve the Physicochemical and Functional Properties of Whey Protein-Based Films to Extend the Shelf Life of Muffins

**DOI:** 10.3390/foods10112672

**Published:** 2021-11-03

**Authors:** Sergio de Jesús Calva-Estrada, Maribel Jimenez-Fernandez, Alba Adriana Vallejo-Cardona, Gustavo Adolfo Castillo-Herrera, Eugenia del Carmen Lugo-Cervantes

**Affiliations:** 1Unidad de Tecnología Alimentaria, Centro de Investigación y Asistencia en Tecnología y Diseño del Estado de Jalisco (CIATEJ) A.C., Camino Arenero 1227, El Bajío, Zapopan C.P. 45019, JAL, Mexico; sce_tau@hotmail.com (S.d.J.C.-E.); gcastillo@ciatej.mx (G.A.C.-H.); 2Centro de Investigación y Desarrollo en Alimentos, Universidad Veracruzana, Av. Doctor Luis Castelazo, Industrial Las Animas, Xalapa Enríquez C.P. 91190, VER, Mexico; 3Consejo Nacional de Ciencia y Tecnología—Unidad de Biotecnología Médica y Farmacéutica, Centro de Investigación y Asistencia en Tecnología y Diseño del Estado de Jalisco (CONACYT-CIATEJ) A.C., Av. Normalistas 800, Colinas de la Normal, Guadalajara C.P. 44270, JAL, Mexico; avallejo@ciatej.mx

**Keywords:** antioxidant capacity, bakery, nanocomposite film, polyphenol content, shelf life, water vapor permeability (WVP)

## Abstract

A novel nanocomposite whey protein-based film with nanoemulsified cocoa liquor (CL) was prepared using one-stage microfluidization to evaluate the emulsion properties and the effect of CL on the film properties by response surface methodology (RSM). The results indicated that the number of cycles by microfluidization had a significant effect (*p* < 0.05) on the particle size and polydispersity of the nanoemulsion, with a polyphenol retention of approximately 83%. CL decreased the solubility (<21.87%) and water vapor permeability (WVP) (<1.57 g mm h^−1^ m^−2^ kPa^−1^) of the film. FTIR analysis indicated that CL modified the secondary protein structure of the whey protein and decreased the mechanical properties of the film. These results demonstrate that applying the film as a coating is feasible and effective to improve the shelf life of bakery products with a high moisture content. This nanocomposite film is easy to produce and has potential applications in the food industry.

## 1. Introduction

Emulsion-based delivery systems have potential applications, including the protection and release of aroma molecules and polyphenols [1]. Oil-in-water (*O/W*) nanoemulsions are heterogeneous systems containing small oil droplets (between 50 and 500 nm) dispersed within an aqueous phase that have attracted particular interest in the food industry for the delivery of bioactive lipophilic components due to their ease of fabrication, small particle size, stability, and enhanced bioavailability of encapsulated compounds [2]. Microfluidization is a dynamic high-pressure shearing with excellent efficiency for the production of nanoemulsions by forcing two immiscible liquids into an interaction chamber via microchannels of a special configuration generating high-velocity microstreams that reduce the size of fat globules by the combined effect of cavitation, shear, and impact [3].

The incorporation of emulsifiers, such as proteins, into nanoemulsions is essential to facilitate the formation of small droplets during homogenization after absorption at the interface, reducing the interfacial tension and protecting droplet aggregation by repulsive steric or electrostatic interactions [4]. Whey proteins are a natural emulsifier widely used in the formation and stability of *O/W* food emulsions [5]. These are highly available and economical dairy industry byproducts composed of a mixture of globular proteins (mainly β-LG, α-LA, and BSA). After heat and microfluidization denaturation, its hydrophobicity is increased, improving its capacity to form an interfacial layer around lipid droplets due to the fast adsorption of whey protein molecules to the droplet surfaces within the homogenizer and promoting its polymerization, constituting a fluid with film-forming capacity that can be consolidated via cold gelation and/or dehydration [6,7].

The development of protein-based films has become the focus of research to produce eco-friendly materials with potential applications as active food packaging, offering their functionality as a vehicle for the incorporation of active agents, such as aromas and antioxidant compounds [8]. Whey protein films are known for forming excellent barriers against oxygen, lipids, and aromas and their interesting mechanical properties. However, their hydrophilic nature and brittle behavior limit their applications as food packaging [9,10]. Recent studies have developed active nanocomposite whey protein films incorporating nanoemulsions to improve their solubility and permeability and to give them antimicrobial and/or antioxidant characteristics by adding bioactive compounds to the dispersed phase. The formulations of these nanocomposite films have generally been developed through two emulsification stages: the first stage involves the formation of the nanoemulsion, and the second stage incorporates the nanoemulsion into the film formulation [11,12,13]. No studies have shown the process combined in a single homogenization step to generate a nanoemulsion with the ability to form a nanocomposite protein-based film.

Cocoa liquor is an important subproduct of cocoa beans (*Theobroma cacao*) produced from dried, fermented, and roasted cocoa beans that are separated from their skins and ground to form cocoa liquor, a product that is rich in polyphenols with antioxidant properties [14]. There is a growing interest in the incorporation of natural antioxidants into active protein films with potential applications in the food industry [9]. However, cocoa liquor has a high fat content (cocoa butter) that causes it to solidify at temperatures under 30 °C [15], and its hydrophobic nature makes it incompatible with aqueous food systems, requiring emulsification technologies, such as microfluidization, and the use of proteins as emulsifying agents to diversify cocoa liquor applications in the food industry.

Therefore, the interest of this work was to optimize the process to obtain a nanocomposite film with cocoa liquor nanoemulsified by one-stage microfluidization, using whey proteins as an emulsifying and film-forming agent, evaluating the effect of the microfluidization conditions on the physicochemical properties and antioxidant capacity of the nanoemulsion, the effect of the cocoa liquor on the properties and microstructure of the film, and its application as an edible coating to improve the shelf life of muffins as a model food.

## 2. Materials and Methods

### 2.1. Materials

The cocoa liquor (CL) sample was acquired from Oaxaca, Mexico. The CL presented a 46.4 ± 1.57% fat content, a 4.70 ± 0.90% moisture content, and a water activity value of 0.39 ± 0.04. Whey protein concentration 80 (WPC) was purchased from Vilher (Empresas Vilher, S.A. de C.V., Guadalajara, Mexico).

### 2.2. Preparation of the Nanoemulsions

Nanoemulsions of CL were prepared with WPC as emulsifier. The protein was solubilized in distilled water to obtain a concentration of 8% (*w*/*w*) and was mixed with 6% (*w*/*w*) glycerol (G). The protein solution was denatured in a Thermomix^®^ vortex operated at 400 rpm and 80 °C for 30 min. Then, the CL was incorporated into the protein solution and homogenized at 400 rpm at 50 °C for 15 min. The pre-emulsion was fed into a microfluidizer to obtain the nanoemulsions under different conditions: homogenization pressure (20,000, 25,000, and 30,000 PSI or in units equivalent to 138.9, 172.4, and 206.8 MPa) and the number of cycles (3, 5, and 7 cycles), varying the concentration of CL (0.5, 2.25, and 4%, *w*/*w*). Each emulsion was evaluated.

#### 2.2.1. Characterization of the Nanoemulsions

The nanoemulsions were characterized in terms of their particle size (recorder as the D(4,3) diameter), polydispersity index (PdI), and zeta potential (ζ-potential) using a Zetasizer Nano ZS (Malvern Instruments, Worcestershire, UK) [2]. The total polyphenol content (TPC) was measured by the Folin–Ciocalteu colorimetric method [16]. The TPC retention (%) of the nanoemulsions (encapsulation efficiency) was calculated with Equation (1):(1)$$\mathrm{TPC}\;retention=\left({\mathrm{TPC}}_{R}÷{\mathrm{TPC}}_{1}\right)\times 100$$
where TPC_R_ is the TPC remaining within the droplets of the nanoemulsion, and TPC_1_ is the content initially added to the emulsion. The antioxidant capacity was evaluated by the DPPH radical inhibition method. All of the results obtained were analyzed using response surface methodology.

##### Effect of the Microfluidization Parameters on the Emulsification of the Cocoa Liquor

Response surface methodology (RSM) was employed to study the effects of the independent variables: cocoa liquor concentration (*x*_1_), homogenization pressure (*x*_2_), and the number of homogenization cycles (*x*_3_), on the particle size (*y*_1_), polydispersity index (*y*_2_), ζ-potential (*y*_3_), polyphenol retention (*y*_4_), and DPPH radical inhibition (*y*_5_) of the nanoemulsions. The experiments were approached using a central composite design of 3^3^. The experiments were generated by Design-Expert version 7.0.0 by Stat-Ease Inc. (Minneapolis, MN, USA). A total of 20 experiments were conducted, including the central point, with five repetitions. The design matrix was created, and the results were statistically analyzed and then converted into a response surface. The designs were evaluated separately based on the influence of the independent variables in the modeling of the nanoemulsion-dependent variables. The experimental data were fitted to a second-order polynomial equation as follows (Equation (2)):(2)$$y=\beta_{0}+\beta_{1}x_{1}+\beta_{2}x_{2}+\beta_{3}x_{3}+\beta_{12}x_{1}x_{2}+\beta_{13}x_{1}x_{3}+\beta_{23}x_{2}x_{3}+\beta_{11}x_{1}^{2}+\beta_{22}x_{2}^{2}+\beta_{33}x_{3}^{2}+\beta_{123}x_{1}x_{2}x_{3}+\beta_{112}x_{1}^{2}x_{2}+\beta_{113}x_{1}^{2}x_{3}+\beta_{122}x_{1}x_{2}^{2}\;$$
where *y* is the dependent response; *x*_1_, *x*_2_, and *x*_3_ are the levels of the independent variables; and *β* is the adjusted parameter in the model. The significance of the estimated regression coefficient for each response variable was assessed using *F*-ratios at a probability (*p*) of 0.05. The adequacy of the response surface models was determined using coefficient of determination (*R*^2^) analysis. Only the significant terms (*p <* 0.05) were included in the model.

### 2.3. Preparation of the Films

Films of the CL and WPC nanoemulsion were formulated. WPC (8% *w*/*w*) and G (used as a plasticizer) were solubilized in distilled water. The protein solution was denatured in a Thermomix^®^ vortexer operated at 400 rpm and 80 °C for 30 min. Then, the cocoa liquor was incorporated into the protein-plasticizer solution and homogenized at 400 rpm at 50 °C for 15 min. The pre-emulsion was processed on the microfluidizer by 5 cycles at 25,000 PSI (172.4 MPa). The films were dried at 25 °C and 50% relative humidity for 24 h and then peeled off for testing. In the formulation, the concentration of cocoa liquor (0, 1, and 2%, *w*/*w*) and plasticizer (5, 6, and 7%, *w*/*w*) of the total film-forming solution were varied. Each edible film formulation was evaluated.

#### 2.3.1. Film Characterization

The solubility of the films was evaluated according to the method of Kurt et al. [17]. The water vapor permeability (WVP) of the films was measured gravimetrically, and the mechanical properties (tensile strength (TS, MPa) and elongation at break (EAB, %) of the films were evaluated using a texture analyzer (TA XT plus, Stable Micro Systems, Godalming, UK) [18]. The results were analyzed using RSM.

##### Effect of the Formulation on the Properties of the Films

RSM was generated to study the effects of the independent variables: cocoa liquor concentration (*x*_1_), and plasticizer concentration (*x*_2_), on the solubility (*y*_1_), WVP (*y*_2_), TS (*y*_3_), and EAB (*y*_4_) of the films. The experiments were approached using a central composite design of 3^2^. A total of 13 experiments were conducted, including the central point, with five repetitions. The response surface was obtained separately for each dependent variable. The experimental data were fitted to a second-order polynomial equation (Equation (2)). The significance of the response surface models was estimated (*p <* 0.05), and adequacy (*R*^2^) was determined.

#### 2.3.2. Microstructure Analysis by CLSM

The autofluorescence of the nanoemulsions was initially evaluated using a Cytation3 Cell Imaging Multi-Mode Reader (BioTek Instruments) to determine the conditions to visualize their fluorescence by confocal laser scanning microscopy (CLSM). The images were acquired using CLSM with a Leica confocal microscope (DM 5500 Q, Wetzlar, Germany) equipped with a 63 × 1.3 oil objective using an excitation wavelength of 405 nm and an amplitude of 420–600 nm.

#### 2.3.3. Fourier Transform Infrared Spectroscopy (FTIR)

FTIR analysis of the cocoa liquor and film samples was performed using Cary 630 Fourier Transform InfraRed (FTIR) spectroscopy with an attenuated total reflectance (ATR) diamond accessory (Agilent Technologies, Mississauga, Canada) at 500–4000 cm^−1^ wavelengths.

### 2.4. Coating Application on the Muffins

Two formulations were selected to evaluate their potential application as a coating for muffins as a model food due to their high moisture content and compatibility. Freshly baked samples were dipped into film solution blends (coating WPC: solution with 8% WPC and 6% G; coating WPC/CL: solution with 8% WPC, 6% G, and 2% CL) for 5 s at room temperature. The excess blend was allowed to drip. Then, the coated muffins were dried at 40 °C for 15 min to solidify the coating.

#### Stability of the Muffins during Storage

The muffins were divided into 3 batches: control muffins (uncoated), WPC muffins (with WPC coating without cocoa liquor), and WPC/CL muffins (with WPC coating and nanoemulsified cocoa liquor). The weight loss and moisture, water activity, and texture parameters of the muffins after they were stored for 5 days at 20 °C and under different controlled conditions of relative humidity (20, 50, and 80% RH) were measured. The muffins were ground and then analyzed with an Aqualab Series 3 water activity meter (Labcell Ltd., Basingstoke, Hants, UK) and dried at 105 °C for 24 h to evaluate their moisture content. The texture of the muffins was evaluated using a TA-XTplus texture analyzer (Stable Micro Systems, Godalming, UK) according to the method of Bartolozzo et al. [19]. Hardness, springiness, and resilience parameters were obtained from the time–force curve.

### 2.5. Statistical Analysis

All experimental values included in the RSM represent the value of the mean of a triplicate analysis (*n* = 3). The standard deviation (SD) from the mean of the experimental values was also calculated. Data analysis was performed using XLSTAT 2019. 2.3 (Addinsoft, Boston, MA, USA).

## 3. Results and Discussion

### 3.1. Effect of Microfluidization Parameters on the Emulsification of Cocoa Liquor

The effects of the microfluidization parameters and the cocoa liquor concentration on the physicochemical properties and the antioxidant capacity of the nanoemulsion were studied (Table 1).

Figure 1 shows the RSM of the effect of the independent variables on the particle size, polydispersity, total polyphenol content, and antioxidant capacity. Figure 1a,b shows the effects of the independent variables on the particle size and PdI after microfluidization at 25,000 PSI (172.4 MPa). Both models were significant (*p* < 0.05), with *R*^2^ values of 0.957 and 0.846, respectively. The factors *x*_1_ (cocoa liquor content) and *x*_2_ (number of cycles) had a strong effect on the particle size and the cocoa liquor concentration (*x*_1_) on the PdI. This was not the case for factor *x*_3_ (homogenization pressure) for both dependent variables. The final models, with the insignificant terms (*p* > 0.05) eliminated, were expressed by the quadratic polynomial Equations (3) and (4):(3)$$\mathrm{Particle}\;\mathrm{size}=424.48-11.73x_{1}-30.45x_{3}+0.007x_{1}x_{2}+0.0047x_{2}x_{3}+1.64x_{3}^{2}$$
(4)$$\mathrm{Polydispersity}\;\mathrm{index}=1.45-0.04x_{1}+0.00003x_{1}x_{2}+0.00002x_{2}x_{3}\;$$

The reduced droplet size of the emulsified fats can improve the mechanical properties of protein-based films [13], but this depends on the operating pressure and the number of cycles that determine the disruptive energy needed to break up the droplets [5]. In this work, the particle size of the emulsions ranged from 206 to 261 nm with a monomodal size distribution (PdI from 0.32 to 0.54) under all microfluidization conditions studied (Figure 1a,b). These results indicate the efficient disruption of the cocoa liquor in small droplets by microfluidization. The droplet diameter decreases with increasing homogenization pressure for whey proteins [1]. The nonsignificant effect of the pressure on the particle size may indicate that the pressures studied are located at the maximum disruptive energy necessary for the formation of the droplets. The particle size decreased with a greater number of cycles. A higher number of cycles ensures that the emulsion droplets undergo a uniform disruptive force [2], increase the surface hydrophobicity of whey proteins, and favor its ability to emulsify smaller fat globules [20]. The combined effect of the previous denaturation by heating and the cycles of microfluidization could have increased the adsorbed denatured whey proteins on the oil droplet surface, favoring smaller droplet sizes [21]. A higher cocoa liquor concentration resulted in a larger droplet diameter under constant homogenizing conditions. This was associated with insufficient protein molecules available to cover the newly formed droplet surfaces when the oily phase increases, with an induced aggregation of droplets and an increase in viscosity that diminished the efficiency of droplet disruption, and/or with a higher rate of collision frequency of the droplets formed [2,4]. The ζ-potentials of the emulsions obtained under the conditions studied were very similar, at −21.83 ± 1.8 mV at neutral pH, so its model was not included. These results might indicate that the droplet interface composition was not altered. The ζ-potential values of the emulsions are related to the net negative charge of the whey protein-coated droplets at neutral pH, in which the protein has more negatively charged carboxyl groups and constitutes a barrier to the droplets approaching electrostatic repulsion, resulting in more stable emulsions [5].

Figure 1c,d indicates the effect of the independent variables on the TPC retention and the antioxidant capacity of the emulsions after microfluidization at 25,000 PSI (172.4 MPa). The models were significant (*p* < 0.05) with *R^2^* values of 0.987 and 0.981, respectively. The cocoa liquor concentration (*x*_1_) and homogenization cycles (*x*_2_) had a strong effect on TPC retention, while only the cocoa liquor concentration (*x*_1_) determined the DPPH radical inhibition of the emulsion. The pressure of homogenization did not affect any of the dependent properties (*p* > 0.05). The final models for TPC retention and antioxidant capacity are represented by Equations (5) and (6):(5)$$\begin{array}{ll} \mathrm{TPC}\;\mathrm{retention}\;\left(\%\right) & \\ & =294.3-109.8x_{1}+9.14x_{3}-2.81x_{1}x_{3}-0.41x_{1}^{2}+0.00003x_{2}^{2}-0.79x_{3}^{2}-0.001x_{1}^{2}x_{2}\\ & +0.69x_{1}^{2}x_{3}-0.00002x_{1}x_{2}^{2} \end{array}$$
(6)$$\mathrm{DPPH}•\;\mathrm{inhibition}\;\left(\%\right)=-31.27+8.46x_{1}+1.79x_{1}^{2}\;$$

The TPC retention ranged from 78 to 99%, and the DPPH radical inhibition ranged from 1 to 35% at a concentration of 100 mg of nanoemulsion per mL. Higher values were observed for higher cocoa liquor contents. Higher TPC retention was observed for fewer homogenization cycles. Despite the decrease in TPC, an effect of microfluidization on the antioxidant capacity of the emulsion was not observed. Similar results were previously observed by Karacam et al. [3]. This may be related to an increase in the temperature that favors the oxidation and/or polymerization of polyphenols, generating compounds with greater antioxidant capacity [16].

The microfluidization conditions of five cycles at 25,000 PSI (172.4 MPa) were selected that would obtain a smaller diameter and size distribution of the nanoemulsion that can improve film properties, favoring a more homogeneous particle distribution throughout the polymer network [22] and preserve to a greater extent the cocoa liquor antioxidant capacity to offer its greatest potential in the development of active packaging material.

### 3.2. Effect of the Formulation on the Properties of the Film

In the second stage, the effect of the formulation treated under the microfluidization conditions previously selected for the nanocomposite film properties was evaluated (Table 2).

Figure 2a shows the effect of the independent variables on the solubility of the films. The model was significant (*p* < 0.05) and adequate (*R*^2^ = 0.927). The cocoa liquor concentration (*x*_1_) had a strong effect on the solubility. The final model for the film solubility is shown below:(7)$$\mathrm{Solubility}\;\left(\%\right)=149.08+31.59x_{1}+3.56x_{2}^{2}$$

The film solubility ranged from 20.99 to 44.36 percent. The low solubility of the films was associated with protein denaturation, which introduces strong interactions and disulfide bonds between protein strands [13]. The films loaded with cocoa liquor showed lower water solubility (*p* < 0.05), possibly related to the formation of a more compact structure due to the greater molecular interactions of the blended film and by the physical effect of the nonpolar components that reduced the interaction of the hydroxyl groups of the protein chains with the water molecules [11,12,13]. Highly water-soluble films can be used as edible coatings, whereas insoluble films can provide a protective function for products with a high moisture content [9]. This suggests that the obtained WPC/cocoa liquor films have potential applications as a coating, with greater effectiveness for products with a low moisture content. Pluta-Kubica et al. [22] reported the solubility of furcellaran/whey protein films incorporated with antioxidant extracts (yerba mate and white tea) to be between 47% and 52%, suggesting that cocoa liquor can be a great alternative for the incorporation of natural antioxidant compounds into whey protein-based films with decreased water solubility.

Figure 2b shows the effect of independent variables on the permeability of the films. The model was significant (*p* < 0.05) with an *R*^2^ of 0.942. The cocoa liquor concentration (*x*_1_) had a strong effect on the permeability of the films. The final model for WVP is represented by Equation (8):(8)$$\mathrm{WVP}\;\left({g\;\mathrm{mm}\;h}^{-1}m^{-2}{\;\mathrm{kPa}}^{-1}\;\right)=7.34+30.74x_{1}-5.94x_{1}^{2}+1.11x_{1}^{2}x_{2}$$

The WVP ranged from 1.27 to 4.21 g mm h^−1^ m^−2^ kPa^−1^. An increase in WVP with an increasing plasticizer amount was observed (Figure 2b). Plasticizers are hygroscopic molecules that have an adverse effect on WVP due to increased molecular mobility and a less dense protein network with a large free volume, permitting greater diffusion of water through the film matrix [10]. Additionally, the results showed a lower permeability of the films with increasing cocoa liquor content. This was probably due to the presence of uniformly dispersed nanoinclusions in the polymer matrix leading to an impermeable and nonporous polymeric structure [10], and by the generation of an interconnecting lipid network within the protein matrix, which increased the hydrophobicity, reducing the water absorption through the film [22]. The permeability of the films obtained is similar to that reported by Talens and Krochta [23] in whey protein-based films incorporated with beeswax or carnouba wax, but with lower permeability compared to the values reported in whey proteins-based films incorporated with fatty acids (>4 g mm h^−1^ m^−2^ kPa^−1^) [24] or with oil of oregano (>8.5 g mm h^−1^ m^−2^ kPa^−1^) [25].

Figure 2c,d shows the effect of the independent variables on the mechanical properties of the films. The models for the tensile strength (TS) and elongation at break (EAB) were significant (*p* < 0.05) with *R2* values of 0.965 and 0.812, respectively. TS was strongly affected by the cocoa liquor (*x*_1_) and plasticizer (*x*_2_) content. Only the cocoa liquor content (*x*_1_) affected the film elasticity. The final models for the mechanical properties are represented by the following Equations (9) and (10):(9)$$\mathrm{Tensile}\;\mathrm{strength}\;\left(\mathrm{MPa}\right)=37.65-3.07x_{1}-11.17x_{2}+0.77x_{1}^{2}+0.87x_{2}^{2}$$
(10)$$\mathrm{Elongation}\;\mathrm{at}\;\mathrm{break}\;\left(\%\right)=-178.64-84.28x_{1}-6.77x_{2}^{2}\;$$

The TS is required to maintain the structural integrity of films. A high TS indicates a high mechanical resistance [6]. The TS of the films ranged from 0.86 to 3.71 MPa. Similar values were previously reported in formulations of whey protein-based films [22]. These results showed that the increase in plasticizer favored a less resistant film structure. A similar phenomenon was reported by Sogut [10], associated with a compatibility that was exceeded between the polymer and the plasticizer, causing destabilization and decreased cohesion by reducing intermolecular forces along the polymer chain [6]. Cocoa liquor reduces the strength of the film. Protein-based films loaded with nanoemulsions showed a considerable reduction in TS attributed to the inability of lipids to form continuous and cohesive matrices [11,22]. These results were similar to the effect observed in whey proteins-based films with the incorporation of beeswax or carnauba (TS ≈ 0.5–3.5 MPa) [23].

The EAB determines the flexibility of the film, which is desirable for easy handling [10]. The EAB was between 7.43% and 27.23%. The values were similar to those reported in recent formulations of whey protein-based films [6,9]. A significant decrease in the EAB of the films was observed after the incorporation of cocoa liquor. The addition of *O/W* nanoemulsions to protein-based film formulations increased their elasticity through a plasticizing effect. However, this effect is strongly dependent on the lipid characteristics and its capacity to interact with the biopolymeric matrix [11,12]. The lipids of the cocoa liquor are solid at room temperature. Therefore, these results may have been caused by the reduced mobility of the protein chains due to the rigid structure conferred by cocoa butter and/or caused by stronger interactions within the cocoa liquor components, such as the polyphenolic compounds, with the protein chains that act as crosslinkers [9].

### 3.3. Microstructure of the Emulsions and Films

For the analysis of the microstructure of the nanoemulsions and films, two formulations were selected comprising a control (without cocoa liquor) and a second with incorporated cocoa liquor. The physical appearance of both the nanoemulsions and selected films are shown in Appendix A, respectively. Figure 3a shows the *O/W* emulsion of cocoa liquor/WPC at a 5% dilution in the bright field, observing two types of main structures, one corresponding to the droplets of the emulsion (marked with white arrows) and other structures that were larger, irregular, and translucent, possibly corresponding to soluble aggregates of whey proteins (WPC) like a gel formed at neutral pH by the combined effect of heat treatment and microfluidization by shelf assembly via hydrophobic attractions and disulfide bond formation [7,21]. This was corroborated by analyzing the fluorescence emission of the same sample at 415 nm (Figure 3b), observing small fluorescent particles (shown in green) corresponding to the emulsified cocoa liquor droplets, not the irregular and large structures that may be associated with protein aggregates formed by an excess of nonadsorbed proteins in the continuous phase that are essential for polymerization and the consequent film formation. This acts as a “glue” between adsorbed protein layers of neighboring droplets, inducing aggregation of the emulsion droplets [26]. This phenomenon is shown in Figure 3c,d, which shows the fluorescence micrograph of the undiluted emulsions with a content of 1 and 2% cocoa liquor, respectively. The presence of the emulsion droplets, and larger fluorescent emulsion droplet aggregates favored by a higher amount and formation of hydrophobic interactions and thiol-disulfide interchanges are observed, resulting in gelation between droplets by the combined effect of heating and microfluidization [26].

Figure 3e–g shows micrographs of the whey protein/cocoa liquor films formed. Figure 3e shows the WPC film without cocoa liquor that did not show autofluorescence, followed by films with 1% and 2% emulsified cocoa liquor (Figure 3f,g, respectively). A homogeneous distribution of the nanoemulsion can be highlighted throughout the polymerized protein matrix, showing in both cases some dark spaces possibly associated with soluble protein aggregates of excess nonadsorbed protein that does not participate in the emulsification of the cocoa liquor but is essential for the formation of the polymeric network and/or is associated with the possible presence of pores.

### 3.4. FTIR Spectroscopy Analysis

The FTIR spectra of the films with and without cocoa liquor were slightly different, indicating that the addition of cocoa liquor affected the film structure (Figure 4). The cocoa liquor spectrum presented important signals at 2916, 2851, and 1740 cm^−1^, which were observed in all film samples with incorporated cocoa liquor. An increase in the intensity of each signal with the increase in the content of the cocoa liquor in the film was observed. The range between 3000 and 2800 cm^−1^ was assigned to the C–H stretching vibrations of alkane groups in the film chains and emulsified fats [10,13,27]. The bands at 2916 and 2851 cm^−1^ can be assigned to the C-H aromatic ring of some molecules present in the cocoa liquor, such as polyphenols [28], and the band at 1740 cm^−1^ may be associated with a C=O stretching vibration likely caused by cocoa liquor triglycerides and carbonyl structures of aldehydes or esters [8,29]. Other changes in important signals different from those for cocoa liquor were observed. The intensity of the 1030 cm^−1^ band showed a slight increase with increasing plasticizer in the film, which is attributed to the C–O stretching characteristic of glycerol [30]. All film samples showed a broad band located at 3000–3600 cm^−1^, corresponding to the stretching vibration of free and bound –OH and –NH groups of the protein matrix [8,10]. Between 1700 and 1500 cm^−1^, high peaks were observed. The peptide bond is normally represented by a C=O stretching vibration at the amide I band (1600–1700 cm^−1^) and the N–H bending vibration and C–N stretching vibration at the amide II band (1500–1600 cm^−1^). A different behavior was found in the amide I band among the film samples. The amide I band is the most useful peak for infrared analysis of secondary protein structures (α-helix, β-sheet, β-turn, and random) of polypeptides or proteins [31]. Four peaks were observed in the amide I region of films with emulsified cocoa liquor, located at 1622, 1634, 1649, and 1690 cm^−1^ (Figure 4). The films without cocoa liquor only presented the first three. These four bands have been previously reported in several studies [32,33]. In all film samples, the highest peak was observed at 1622 cm^−1^. This signal is assigned to intermolecular β-sheets resulting from aggregations during network formation. The bands at 1634 also indicate the formation of intramolecular β-sheet structures, the bands at 1649 cm^−1^ correspond to α-helices, and 1690 cm^−1^ corresponds to anti-parallel β-sheets or β-turns that were only present in the films with higher cocoa liquor content regardless of the amount of plasticizer. Therefore, the addition of cocoa liquor to the film formulation is accomplished by modifying the secondary structure of whey proteins, favoring the unfolded proteins (random coil) folded into α-helix and β-turn structures more than the β-sheet during the rebuilding of the secondary structure in the emulsification and film formation process [32]. The major α-helix and β-sheet structures formed with the incorporation of cocoa liquor were corroborated with their respective spectral amide III regions arising at 1295 and 1220 cm^−1^, respectively [27]. A shift of the band at 1621 cm^−1^ to 1627 cm^−1^ was observed with the incorporation of the cocoa liquor in the films (Figure 4). The diminished position of the band indicates a strengthening of the intermolecular hydrogen bonds between β-sheets and it is associated with the effect of the α-helix structure formed [33]. The molecular protein conformation can be related to the mechanical properties of the films. These results indicate that cocoa liquor favors β-turns and β-sheet folds that contribute to diminishing the flexibility and tensile strength of the films, possibly because the more ordered β-sheets produce a stronger protein network through hydrogen bonding [31].

### 3.5. Coating Application on Muffins and Storage Study

Images of the appearance of the muffins with the different coatings studied are shown in Appendix A, observing that the application of the coatings was primarily associated with a brighter surface compared to the uncoated muffins. The effect of the coatings on the quality properties of the muffins during their shelf life under the studied conditions is shown in Figure 5. The kinetic parameters calculated from the charts (parameter vs. time) are shown in Table 3. The results indicate that under conditions of relative humidity (RH) of 50 and 20%, there was a significant decrease in the weight (Figure 5a), moisture content (Figure 5b), and water activity (a_w_) (Figure 5c) in the muffins. At 80% RH, the properties of the muffins (coated and uncoated) did not change significantly during storage. This can indicate that the relative humidity of 80% is close to the moisture content of the product, and a higher difference between the moisture content of the product with respect to that of the surrounding medium, observed at a lower relative humidity, favors a greater transfer amount of moisture from the food to the outside environment, which leads to the product losing weight. Under conditions of intermediate relative humidity (50%) and low relative humidity (20%), a lower loss of humidity and a_w_ was evident in the coated muffins compared to the control. These results suggest that the coating on the surface of the muffins hindered moisture loss and improved their water holding capacity during storage [34].

Texture (hardness, resilience, and springiness) is one of the main characteristics of bakery products, and the loss of the desired texture results in a shorter shelf life. The firmness or hardness of the muffins constitutes the most important texture parameter due to its strong correlation with the consumer’s perception of the freshness of the product [19]. Therefore, the increase in hardness may be associated with the loss of moisture of the product and its freshness. Considering the results obtained under conditions of low relative humidity (20%), it can be observed that the muffins with the WPC/CL coating proved to be more effective relative to the samples with the WPC coating and control during storage, presenting a *k* value significantly lower for weight loss (−7.42 ± 0.44 (g/100 g)/day), moisture (−16.83 ± 0.52 (g/100 g)/day), and a_w_ (−0.080 ± 0.00 a_w_/day) and a smaller increase in hardness (8.87 ± 1.45 N/day).

## 4. Conclusions

The results showed that a greater number of homogenization cycles favored the hydrophobicity of the protein, decreasing the particle size and polydispersity of the emulsion. Microfluidization did not affect the ζ-potential and had a minimal impact on the antioxidant capacity of the cocoa liquor, allowing for the retention of a high percentage of polyphenolic compounds. Five microfluidization cycles with a pressure of 25,000 PSI (172.4 MPa) were the conditions that favored better physicochemical properties in the emulsion while also retaining its antioxidant capacity. The plasticizer content significantly decreased the strength of the film, and the cocoa liquor content decreased its water solubility and water vapor permeability. The latter was associated with modifications of the secondary protein structure influenced by the incorporation of cocoa liquor during the emulsification and film formation process as evidenced by the FTIR analysis. The results demonstrate the feasibility and effectiveness of applying WPC-based coatings to extend the shelf life of bakery products with a high moisture content, such as muffins, and the contribution of cocoa liquor nanoparticles in decreasing the permeability and loss of weight and moisture of the product. This novel active nanocomposite whey protein-based film with cocoa liquor nanoparticles may have potential applications in the food industry because its characteristics can be mainly used as a coating to control moisture transfer and prevent oxidation due to its antioxidant properties.

## Figures and Tables

**Figure 1 foods-10-02672-f001:**
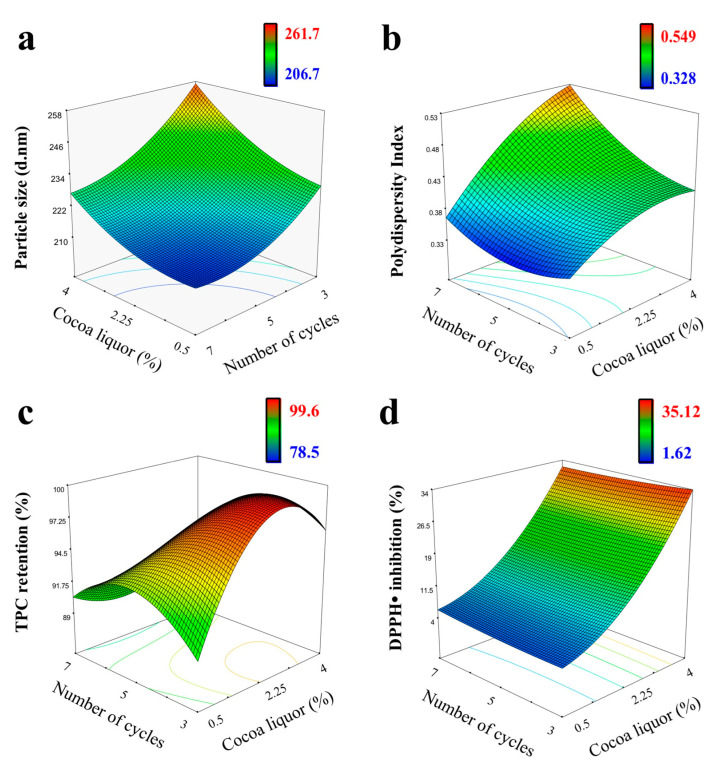
Three-dimensional response surface plot for the interactive effect of independent variables (cocoa liquor concentration and homogenization cycles) on: (**a**) particle size; (**b**) polydispersity index; (**c**) polyphenols retention; and (**d**) DPPH radical inhibition of the nanoemulsion with 8% of whey protein after microfluidization at 25,000 PSI (172.4 MPa).

**Figure 2 foods-10-02672-f002:**
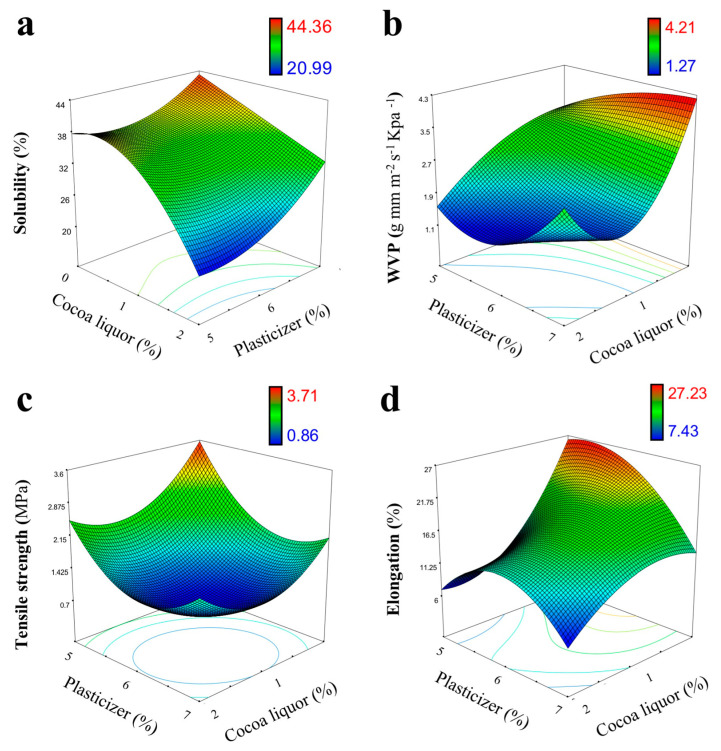
Three-dimensional response surface plot for the interactive effect of independent variables (cocoa liquor concentration and plasticizer concentration) on: (**a**) solubility; (**b**) water vapor permeability; (**c**) tensile strength; and (**d**) elongation at break of the whey protein-based films with 8% protein content.

**Figure 3 foods-10-02672-f003:**
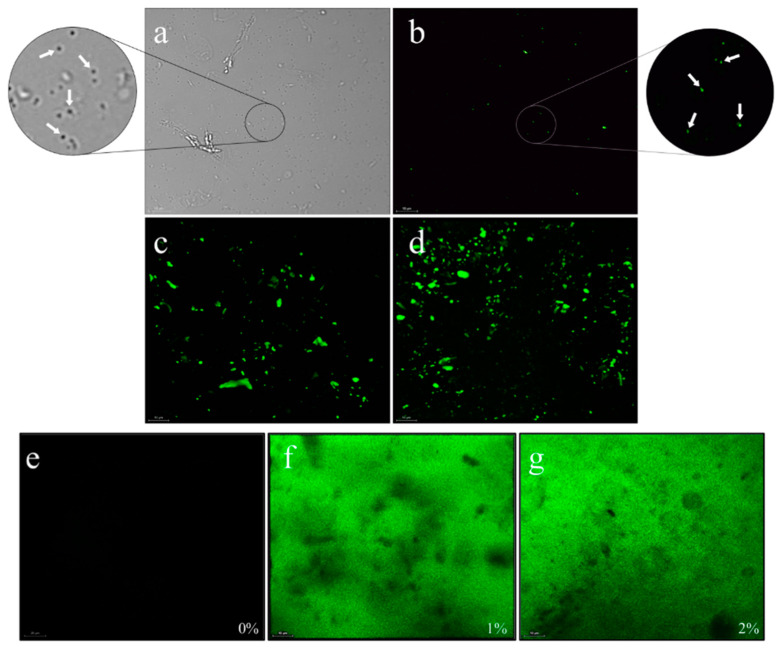
Microstructure of the (**a**) nanoemulsion with 8% Whey Protein Concentrate (WPC), 6% Glycerol (G), and 2% cocoa liquor (CL) diluted 5% in the bright field; (**b**) nanoemulsion with 8% WPC, 6% G, and 2% CL diluted 5%; (**c**) nanoemulsion with 8% WPC, 6% G, and 1% CL; (**d**) nanoemulsion with 8% WPC, 6% G, and 2% CL; (**e**) film with 8% WPC, 6% G without CL; (**f**) film with 8% WPC, 6% G, and 1% CL; and (**g**) film with 8% WPC, 6% G, and 2% CL observed by fluorescence with a Confocal Laser Scanning Microscope (CLSM). White arrows show emulsion droplets from an amplified field fraction. The scale bar represents a length of 10 µm. The green regions represent the autofluorescence of the cocoa liquor polyphenols.

**Figure 4 foods-10-02672-f004:**
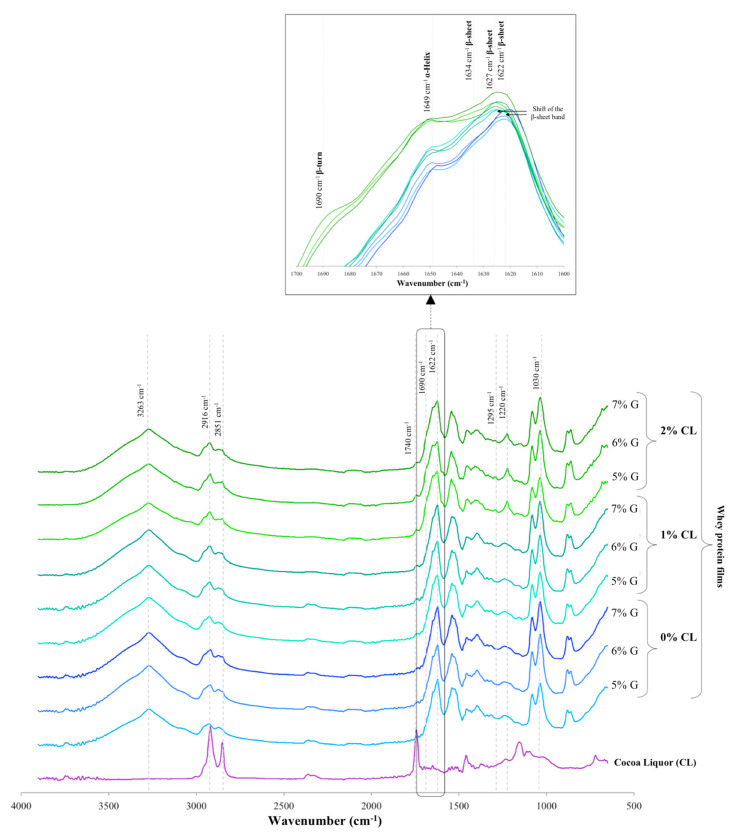
Comparison of FTIR absorbance spectra of the cocoa liquor and whey protein films with different cocoa liquor (CL) (0, 1, and 2%) and glycerol (G) (5, 6, and 7%) contents, indicating the most relevant absorption signals, and the signals associated with the different secondary conformations (α-helix, β-sheet, and β-turn) in the amide I region (1700–1600 cm^−1^) of the proteins in the films.

**Figure 5 foods-10-02672-f005:**
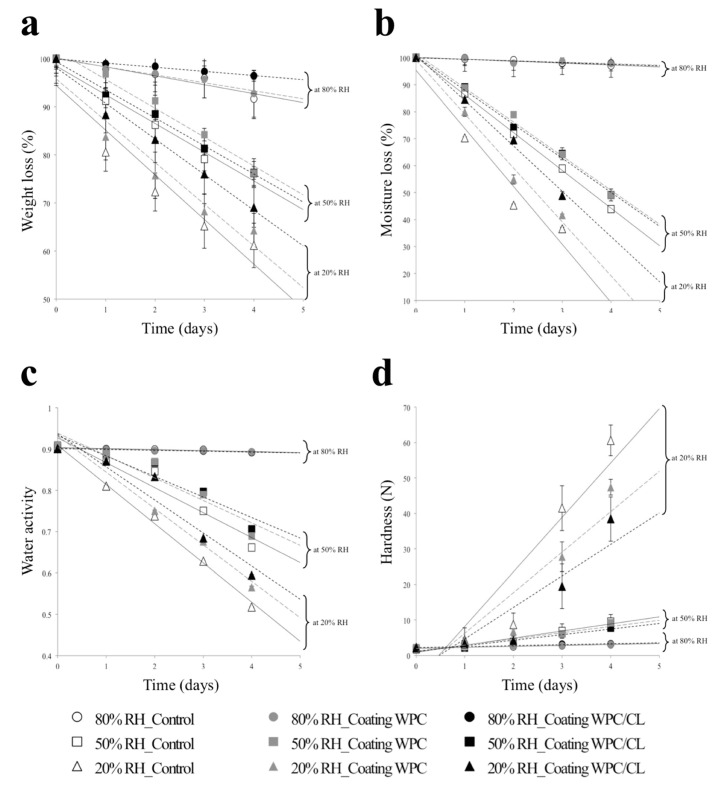
Effect of the coatings on the stability parameters of the muffins stored at 20 °C and different conditions of relative humidity (20, 50, and 80%): (**a**) weight; (**b**) moisture content; (**c**) water activity; and (**d**) hardness.

**Table 1 foods-10-02672-t001:** Central composite experimental design, presenting the effect of the independent variables on the physicochemical properties and antioxidant capacity of the nanoemulsion. The total number of runs was generated by Design Expert version 7.0.0 by Stat-Ease Inc. (Minneapolis, MN, USA).

	Independent Variable				Response			
Run	Cocoa Liquor	Homogenization Pressure	Number of Cycles	Particle Size	Polydispersity Index	ζ-Potential	TPC Retention	DPPH^•^ Inhibition
	(*x*_1_)	(*x*_2_)	(*x*_3_)	(*y*_1_)	(*y*_2_)	(*y*_3_)	(*y*_4_)	(*y*_5_)
	%	PSI (Equivalent in MPa)		*d* nm		mV	%	%
1	4.00	25,000 (172.4)	5	232.00	0.42	−22.30	98.73	28.38
2	5.00	20,000 (138.9)	7	209.50	0.36	−22.30	98.54	3.39
3	0.50	30,000 (206.8)	3	210.26	0.38	−21.60	93.05	5.44
4	2.25	25,000 (172.4)	5	217.33	0.39	−22.70	96.51	15.01
5	2.25	25,000 (172.4)	5	212.13	0.37	−22.90	95.12	13.50
6	0.50	25,000 (172.4)	5	212.13	0.33	−22.83	93.99	8.04
7	2.25	30,000 (206.8)	5	213.10	0.40	−22.83	95.36	12.61
8	2.25	25,000 (172.4)	5	213.38	0.38	−21.90	97.20	14.32
9	4.00	30,000 (206.8)	7	251.43	0.55	−20.03	78.50	35.12
10	4.00	30,000 (206.8)	3	261.76	0.45	−20.30	83.03	34.29
11	2.25	25,000 (172.4)	5	219.24	0.39	−23.30	97.45	14.20
12	0.50	20,000 (138.9)	3	227.86	0.40	−22.10	97.76	1.62
13	2.25	25,000 (172.4)	5	221.33	0.39	−22.76	96.51	13.83
14	2.25	20,000 (138.9)	5	226.43	0.45	−22.90	97.13	11.24
15	0.50	30,000 (206.8)	7	206.76	0.36	−22.33	94.19	5.10
16	2.25	25,000 (172.4)	5	214.43	0.40	−22.63	94.95	13.83
17	2.25	25,000 (172.4)	7	215.96	0.42	−22.10	90.09	13.90
18	4.00	20,000 (138.9)	7	225.70	0.38	−22.23	87.85	29.38
19	2.25	25,000 (172.4)	3	213.36	0.42	−23.63	99.64	12.28
20	4.00	20,000 (138.9)	3	258.66	0.41	−20.16	89.39	34.79

*x*: represents the independent variables plotted on the *x*-axis; *y*: represents the response plotted on the *y*-axis. Response is the mean value of three replicates. TPC = Total Polyphenol Content; DPPH^•^ = 1-DiPhenyl-2-PicrylHydrazil radical; PSI: Pounds per Square Inch unit; MPa: MegaPascal unit.

**Table 2 foods-10-02672-t002:** Central composite experimental design, presenting the effect of the independent variables on the film properties. The total number of runs was generated by Design Expert version 7.0.0 by Stat-Ease Inc.

	Independent Variable		Response			
Run	Cocoa Liquor	Plasticizer	Solubility	WVP	TS	EAB
	(*x*_1_)	(*x*_2_)	(*y*_1_)	(*y*_2_)	(*y*_3_)	(*y*_4_)
	%	%	*%*	g mm h^−1^ m^−2^ Kpa^−1^	MPa	%
1	1.00	6.00	35.28	1.67	0.94	24.00
2	1.00	7.00	35.82	1.65	1.17	11.71
3	2.00	5.00	21.87	1.57	2.69	8.06
4	2.00	7.00	33.70	2.72	1.94	7.43
5	0.00	6.00	34.88	3.97	1.82	22.76
6	1.00	5.00	34.07	2.80	1.96	8.74
7	1.00	6.00	32.97	2.46	0.86	16.16
8	2.00	6.00	20.99	1.27	1.10	11.54
9	1.00	6.00	36.13	1.60	1.13	25.53
10	1.00	6.00	34.15	2.08	0.93	20.17
11	0.00	5.00	38.94	3.21	3.71	27.23
12	1.00	6.00	33.39	1.49	0.97	17.02
13	0.00	7.00	44.36	4.21	2.15	14.11

*x*: represents the independent variables plotted on the *x*-axis; *y*: represents the response plotted on the *y*-axis. Response is the mean value of three replicates. WPV: Water Vapor Permeability; TS: Tensile Strength; EAB: Elongation At Break.

**Table 3 foods-10-02672-t003:** Kinetics parameters from the quality properties of the stored muffins at 20 °C and different relative humidity (20, 50, and 80%) for 5 days.

Sample	RH (%)	Weight Loss		Moisture Loss		Water Activity		Hardness	
		*r^2^*	*k* (g/100 g)/day	*r^2^*	*k* (g/100 g)/day	*r^2^*	*k* a_w_/day	*r^2^*	*k* N/day
Control	80	0.9106	−1.86 ± 0.08 ^a^	0.9632	−0.72 ± 0.01 ^a^	0.8834	−0.002 ± 0.000 ^a^	0.9945	0.27 ± 0.04 ^a^
Coating WPC	80	0.9585	−1.66 ± 0.09 ^a^	0.8068	−0.56 ± 0.03 ^b^	0.9248	−0.002 ± 0.000 ^a^	0.9441	0.29 ± 0.02 ^a^
Coating WPC/CL	80	0.9905	−0.86 ± 0.01 ^b^	0.8255	−0.60 ± 0.02 ^b^	0.9343	−0.002 ± 0.000 ^a^	0.8932	0.28 ± 0.04 ^a^
Control	50	0.977	−5.98 ± 0.19 ^a^	0.9994	−14.01 ± 0.11 ^a^	0.9291	−0.060 ± 0.00 ^a^	0.9457	2.00 ± 0.42 ^a^
Coating WPC	50	0.9757	−6.01 ± 0.10 ^a^	0.9919	−12.66 ± 0.44 ^b^	0.8913	−0.054 ± 0.00 ^b^	0.8683	1.75 ± 0.24 ^a^
Coating WPC/CL	50	0.9924	−5.84 ± 0.22 ^a^	0.9957	−12.63 ± 0.17 ^b^	0.901	−0.050 ± 0.00 ^c^	0.9171	1.58 ± 0.16 ^a^
Control	20	0.9139	−9.30 ± 0.70 ^a^	0.9525	−21.51 ± 0.29 ^a^	0.9961	−0.095 ± 0.00 ^a^	0.8638	15.35 ± 1.09 ^a^
Coating WPC	20	0.9396	−8.69 ± 0.48 ^a^	0.9871	−20.00 ± 0.19 ^b^	0.9652	−0.088 ± 0.00 ^b^	0.8529	11.40 ± 0.53 ^b^
Coating WPC/CL	20	0.9832	−7.42 ± 0.44 ^b^	0.994	−16.83 ± 0.52 ^c^	0.9235	−0.080 ± 0.00 ^c^	0.8075	8.87 ± 1.45 ^c^

Control = uncoated muffins; Coating of Whey Protein Concentrate (WPC) = muffins with coating solution 8% WPC; Coating of WPC and Cocoa Liquor (CL) (WPC/CL) = muffins with coating solution 8% WPC and 2% CL. RH = Relative Humidity; a_w_ = water activity; *k* = rate constant; N = Newtons. Data are expressed as means (*n* = 5) ± SD. Different letter superscripts within a column indicate statistically significant differences (*p* < 0.05) per relative humidity condition.

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
