# Peer review of "Cocoa Nanoparticles to Improve the Physicochemical and Functional Properties of Whey Protein-Based Films to Extend the Shelf Life of Muffins"

_foods, 2021, doi:10.3390/foods10112672_

Round 1

Reviewer 1 Report

In my opinion, this research is very interesting and the manuscript has been well written. 

However, I have some doubts about the target/aim of the present study, which, in reality, is represented by two parts, one of which (the second one) was not well done.

Indeed authors deeply treated the part relative to the study of the chemical/phisical characteristics and performances (except for the phenolic composition) of the film with and without cocoa liquor but they did not reserve the same importance/accuracy for the shelf life study. In particular,  they evaluated a very limited number of parameters for having a complete response of the goodness of the different films in preserving the quality of a bakery product like a muffin, strongly subjected to the storage time and conditions  which can promote  very important modifications in its sensory attributes (which are not only by tactile sensations). Authors, in fact, only investigated strumental parametrs indirectly correlable with the loss of sofficitiy and the increase of the hardness. 
Other human sensors involved, such as appereance, odour, taste, flavour and other tactile sensations which can be strongly modfied over storage and which can be modifed by using those  films, were completely ignorated. So, why did not the authors carry out a proper sensory analyisis (such as a QDA) on the muffins during the shlef life? I think that is absolutely necessary in these kind of studies. Furhermore, I also found that the chemical characteristics of the cocoa oil and the films  were not completely evaluated. In particular, their volatile and fatty acid composition was not investigated. 

In this regard, both volatiles and fatty acids might be correlated with specific olfactory (volatiles) and tactile viscosity, fluidity) sensory aspects (fatty acids). Again, I have to underline the necessity of a sensory analysis study, also for covering the lacks of a complete screening of the chemical composition of the cocoa oil.

Moreover, authors evaluated the phenolic composition of the cocoa oil by Folin Ciocalteau method which, as well known, implies the use of certain reagents which react, not only with the phenols but also with other substances, overstimating the relative results. However, given the importance of the phenolic composition of the cocoa oil in this research, which probably could have a positive effect against the oxidative phenomena (other not invetsigate aspect) over storage, a quali/quantitative analysis of the single phenolic fractions  of the film should be done.

In conlusion, authors should provide, at least, sensory analysis of the muffin for justifing the inclusion of the seceond part of this study. Otherwise they should focus the entire manuscript (title and introduction included) on the first part of the study. However, a more complete chemical evaluation (a quali/quatitative evaluation of volatile and phenolic fractions) of the films, should be done. In fact, the film characterization (as the authors indicated in materials and methods)  cannot be represented only by a total phenols analysis!!!

Author Response

Author response:

We appreciate the valuable comments made by the reviewer.  We agree that it would be interesting to go even deeper into our study of shelf life. However, we consider that the results presented allow us to show the benefits of using the film developed in the preservation of the quality of the product used as a model based on the monitoring of base parameters evaluated in studies of the same approach. The conditions to which the samples were subjected during the shelf life study were selected in order to demonstrate their behavior under different humidity conditions of the environment, which affects the stability and therefore the efficiency and functionality of this type of coatings. Thus, allowing the identification of ranges of environmental conditions that maintain the functionality of the film based on the food matrix on which it is adhered/formed.

We also agree that the sensory evaluation of the product would provide valuable information on acceptability and in the identification of changes in the product from the perception of the same consumer, but that they are not part of the objective of this research, since through our results We want to demonstrate the application potential of the film developed by opening the possibility of studying the improvement of the application process or testing its versatility in a model food other than the one proposed, which is not limiting at all.

We know of the great importance that many of the other main components of cocoa liquor can have, such as its profile of fatty acids, the profile of phenolic compounds, including its alkaloids theobromine and caffeine, and its complex volatile composition, and that together, They offer characteristics that undoubtedly should show a sensory perception of great interest and possibly to the benefit of the benefits of our proposed coating, and that can only be given in value after carrying out a sensory analysis of it. However, the proposed study was focused on the optimization and improvement of the process that allowed to establish the conditions to obtain the greatest possible interaction between the proposed system (protein-cocoa liquor) that had the potential for the development of the film and to study how these conditions affected one of the important characteristics of the cocoa liquor that we were mainly interested in maintaining to the greatest extent, the antioxidant load of the cocoa liquor as a whole, rather than a particular compound, by means of a rapid and economically accessible analysis, but to at the same time reliable and of scientific validity for the monitoring of processes, due to the multiple trials that included the study.

Reviewer 2 Report

It's highly recommended to evaluate the mechanical properties of the film. 

Compare the results with similar studies. 

Evaluate the degradation rate of the film. 

It is recommended to add the film image.  

It is recommended to use the DSC test. 

Author Response

Response to Reviewer 2 Comments on Manuscript ID foods-1386962

It's highly recommended to evaluate the mechanical properties of the film. 

Author response: Mechanical properties of the film (Tensile strength and elongation at break) were included in Table 2.

Compare the results with similar studies. 

Author response: Some comparisons of our results with other similar studies were included.

Evaluate the degradation rate of the film. 

Author response: We appreciate your observation. The degradation rate of the film was not evaluated, but we consider that the percentage of solubility included in table 2 allows us to estimate the differences in the degradation of the film.

It is recommended to add the film image.  

Author response: We appreciate your recommendation. Images corresponding to the appearance of the emulsions, films, and coated muffins were included as supplementary material.

It is recommended to use the DSC test. 

Author response: We agree with the reviewer that thermal analysis, as well as other studies, would be interesting and increase knowledge about coatings. However, the thermal analysis by DSC was not considered from the beginning of the experiment, so the inclusion of this study would imply the establishment of different storage conditions and unfortunately at this time, we do not have the resources or the equipment to carry it out. However, we consider that the evaluated properties allow an adequate assessment of the capacity and properties conferred by the film in the useful life of the products developed.

Reviewer 3 Report

1. psi is not an SI unit, please correct it throughout the manuscript.

2. Please use the math equation editor to correctly edit math expressions. The manuscript does not contain the characteristics of nemulsions. Basic parameters were not specified: droplet diameter distribution, d32, d34 etc. Without these parameters, nothing can be said about the properties of the emulsion. 

Author Response

Respuesta a los comentarios del revisor 3 sobre el manuscrito ID foods-1386962

  1. psi no es una unidad SI, corríjala en todo el manuscrito.

Respuesta del autor: Gracias por la observación. Los valores de presión de homogeneización en PSI se referenciaron en su valor correspondiente en MPa, una unidad de presión del sistema internacional.

  1. Utilice el editor de ecuaciones matemáticas para editar correctamente las expresiones matemáticas.

Respuesta del autor: Gracias por la observación. Se aplicó el cambio.

El manuscrito no contiene las características de las nanomulsiones. No se especificaron los parámetros básicos: distribución del diámetro de las gotas, d32, d34, etc. Sin estos parámetros, no se puede decir nada sobre las propiedades de la emulsión. 

Respuesta del autor: Gracias por la observación. Esta información se especificó en la descripción del método.

Round 2

Reviewer 2 Report

The authors address all comments and I recommend publishing the article. 

Author Response

Dear reviewer

On behalf of all the authors, we cordially thank their valuable contributions and helpful comments for the improvement of our manuscript. And we tell you that we have also carried out a detailed grammar review of the document.

For your support, thank you.

Sincerely,

Eugenia Lugo